# Selection on heritable social network positions is context-dependent in *Drosophila melanogaster*

Eric Wesley Wice 1✉ & Julia Barbara Saltz 1

Social group structure is highly variable and can be important for nearly every aspect of behavior and its fitness consequences. Group structure can be modeled using social network analysis, but we know little about the evolutionary factors shaping and maintaining variation in how individuals are embedded within their networks (i.e., network position). While network position is a pervasive target of selection, it remains unclear whether network position is heritable and can respond to selection. Furthermore, it is unclear how environmental factors interact with genotypic effects on network positions, or how environmental factors shape selection on heritable network structure. Here we show multiple measures of social network position are heritable, using replicate genotypes and replicate social groups of *Drosophila melanogaster* flies. Our results indicate genotypic differences in network position are largely robust to changes in the environment flies experience, though some measures of network position do vary across environments. We also show selection on multiple network position metrics depends on the environmental context they are expressed in, laying the groundwork for better understanding how spatio-temporal variation in selection contributes to the evolution of variable social group structure.

---

[1] Department of Biosciences, Rice University, Houston, TX, USA. ✉email: eric.wesley.wice@gmail.com

From classic examples of eusocial animals, to species that are commonly thought of as solitary, the diversity of social structures evident in nature fascinates biologists and naturalists. Characteristics of social group structure emerge from the patterning and organization of social interactions occurring between members of a social group, which are commonly modeled using social network analysis[1–3]. How individuals are embedded within their social groups, termed "social network position", has been demonstrated to influence many critical aspects of an individual's life (mating opportunities[4,5], risk of disease[1,6], access to social information[7,8], foraging potential[8,9], etc.). Despite this importance and pervasive evidence that social network positions are under selection, we still know relatively little about the underlying causes of variation in social network positions and group structure. Such an understanding is necessary for identifying ways that social structures may form and evolve, provoking recent calls for studies investigating the mechanisms underlying social network structure[10–12]. Potential mechanisms that could give rise to variation in social network structure include the genotypes of individuals comprising a social group, as well as the environmental conditions in which social interactions occur[13,14]. However, very few studies have investigated whether social network positions of individuals within a group are heritable[10,11,15–20], or how the structure of a social network varies across environments[12,15,21]. Furthermore, we know even less about how selection shapes heritable variation in social network structure over generations[22].

One of the challenges of studying the genetic basis of and how selection acts on social network positions is that network position is not inherent to the individual, and is instead an extended phenotype dependent on the direct and indirect interactions of conspecifics in an individual's social group[23,24]. Thus, to measure the genetic basis of social network position, one has to measure focal individuals of known genotype within a replicate social group comprised of the same conspecifics[25]. Repeatedly measuring the same social group in the wild presents additional challenges, as prior environmental and social experiences can induce plasticity and variation in social network positions[6,26,27], and/or reinforce individual differences in social network positions[20,26,28–31]. These empirical challenges leave gaps in our knowledge about whether social network positions are heritable traits that can respond to selection, as no prior study has estimated the heritability of individuals' network positions while also controlling for the genotypes of their social partners and prior experience. Furthermore, no prior study has investigated the heritability of network positions in conjunction with environmental variation, and no prior study has addressed how selection acts on network positions in tandem with known genotypic and environmental influences. Studying the causes of variation in social network structure, including how genotypic and environmental influences align with patterns of selection on social network positions, is crucial for understanding how diverse patterns of social group structures form and evolve[15,22].

In the present study, we identify how genetic, environmental, and genotype-by-environment interaction effects contribute to variation and plasticity in commonly studied social network positions. We also address how selection acts on social network positions, across environmental contexts, to shape and adaptively maintain genetic variation in network structure[32–36].

*Drosophila melanogaster* flies are an ideal study system for investigating the genetic and environmental causes of social network positions, in tandem with the fitness consequences of social network positions. Flies form non-random social networks[7,13] and spend the majority of their lives living and socially interacting on nutritionally-variable rotting fruit environments[37]. We can control and replicate the genotypes and environments of all individuals within a social group, allowing us to create replicate social groups with genetically identical composition and measure their social structure across variable environments[38]. We created 98 replicate social groups of flies. Each social group consisted of 20 unrelated, heterozygous genotypes bred from the *Drosophila* Genetic Reference Panel (DGRP) (10 males and 10 females per replicate group)[38]. Each replicate social group was placed on one of five nutritional environments varying in either protein-to-carbohydrate ratio or caloric concentration of the food, both of which have been shown to affect various components of fly behavior and fitness[39–41]. Social groups were video recorded twice over the course of two days, and social networks were generated using automated motion tracking software, resulting in 56 weighted and directed networks comprising over 600,000 s of social interactions[42]. We analyzed the five most commonly studied social network positions for each individual within a social group[3]: Instrength and outstrength—the amount of time other individuals spend socially engaging with a focal individual, and the amount of time a focal individual spends socially engaging with other individuals, respectively; Clustering coefficient—how interconnected a focal individual's direct social partners are to one another (i.e., cliquishness); Betweenness centrality—the number of shortest paths between any two individuals that transverse a focal individual; and Eigenvector centrality—how critical a focal individual is to the overall structure of the group based on the strength of its direct social connections, the strength of its partners' connections, the strength of its partners' partners' connections, etc.[1,2]. We also measured multiple metrics of fitness for each individual, allowing us to estimate the strength and direction of selection on social network positions. These fitness measures include the total number of observed matings and the latency to mate for males, and the lifetime reproductive success and lifespan of females (Supplementary Fig. 1).

In this work, we find that genotype significantly affects all five measures of social network position, with low to moderate estimates of broad-sense heritability. We also find effects of sex, genotype-by-environment interactions, and sex-by-environment interactions on many measures of network position. Selection on network positions is limited to male flies, and the strength and direction of selection on measures of network position varies depending on the nutritional environment. Spatio-temporal variation in selection, combined with heritable variation for traits under selection, shapes and maintains genetic variation over generations[32,36]. These findings not only suggest that the structure of social groups can respond to selection and evolve, they also shed light on how variation in social structure may evolve under variable environmental conditions.

## Results

**Social network positions**. We discovered that an individual's genotype was a significant predictor for all five network positions analyzed (likelihood ratio tests: instrength, LRT = 22.713, $P_R <$ 0.001; outstrength, LRT = 24.13, $P_R < 0.001$; clustering coefficient, LRT = 54.726, $P_R < 0.001$; betweenness centrality, LRT = 1086, $P_R = 0.022$; and eigenvector centrality, LRT = 24.348, $P_R < 0.001$; Fig. 1). Estimates of broad sense heritability ($H^2$), defined as the extent to which genotypic differences explain variation in a phenotype[43], ranged from 2.4 - 16.6% for the five network positions analyzed (instrength, $H^2 = 0.024$; outstrength, $H^2 = 0.025$; clustering coefficient, $H^2 = 0.050$; betweenness centrality, $H^2 = 0.166$; and eigenvector centrality, $H^2 = 0.042$). We also found significant differences between the sexes in four of the five network positions analyzed (type III Wald $\chi^2$ tests: instrength, $\chi^2 = 8.072$, $P_R = 0.002$; outstrength, $\chi^2 = 10.108$, $P_R = 0.001$;

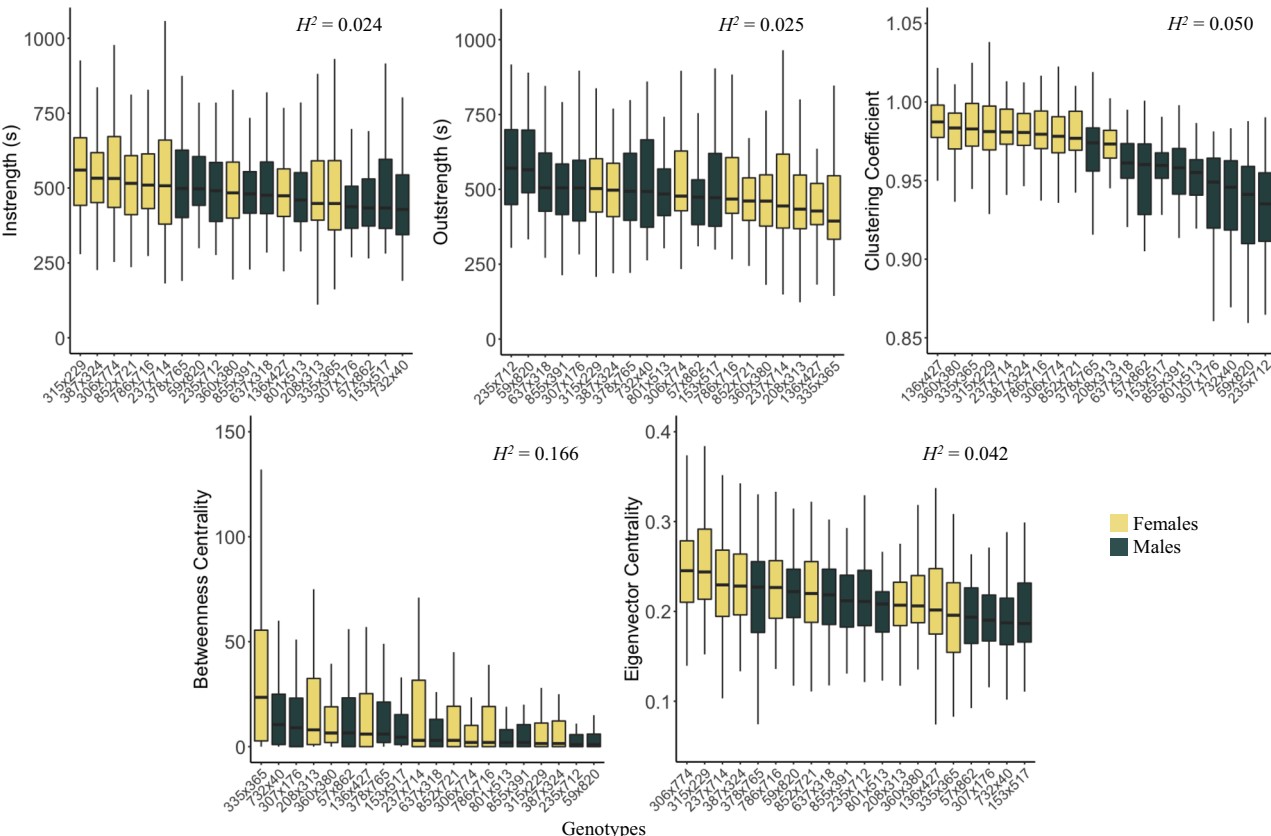

**Fig. 1 Genotypic effects on network position.** Twenty heterozygous *Drosophila melanogaster* genotypes significantly differed in five commonly studied measures of social network position (all LRT, $P_R < 0.05$). Broad-sense heritability estimates ($H^2$) for each network position are presented in each panel. Boxplots present the medians (horizontal lines), interquartile ranges (boxes), and values within ± 1.5x IQR (whiskers) for each genotype ($n$ for each genotype = 56). Genotypes are ordered in each panel by their median value for each network position. Genotype labels are presented as the maternal parent genotype number, crossed with ("x") the paternal genotype number. Genotype numbers are arbitrary and unindicative of any similarities/differences between genotypes. The sex of each genotype is indicated by the color-fill of the boxplots (yellow = females, navy = males). Source data are provided as a Source Data file.

clustering coefficient, $\chi^2 = 106.832$, $P_R < 0.001$; and eigenvector centrality, $\chi^2 = 9.248$, $P_R = 0.006$; Fig. 1), but not for betweenness centrality ($\chi^2 = 6.208$, $P_R = 0.683$; Fig. 1). As expected, females received more social interactions (instrength) than males, and males initiated more social interactions (outstrength) than females. Males also tended to be more central to the overall structure of the social group (eigenvector centrality), while females tended to be more cliquish (clustering coefficient) compared to males (Fig. 1). We saw no effects of genotype-by-environment interactions on the social network positions of instrength, outstrength, betweenness centrality, and eigenvector centrality; meaning that genotypic differences in these measures of network position remained constant across the nutritional contexts (all $P_R > 0.05$; Supplementary Table 1). However, we did find evidence of genotype-by-environment interactions influencing how cliquish (clustering coefficient) individuals were (LRT = 5.000, $P_R = 0.007$) across nutritional environments that varied in both protein:carbohydrate ratio (LRT = 1.843, $P_R = 0.001$) and caloric concentration (LRT = 1.047, $P_R = 0.028$). When selection acts on traits that are influenced by genotype-by-environment interactions, the genotype(s) with the highest relative fitness depends on the environmental context in which the trait is expressed. Thus, genotype-by-environment interactions are one of the primary mechanisms invoked to explain the persistence of genetic variation in traits that are under selection[33–35]. Our findings of genotype-by-environment interactions for how cliquish individuals are (clustering coefficient) provides a

potential mechanism for explaining the maintenance of genetic diversity in this trait. We also find sex-by-environment interactions for the network positions of outstrength and clustering coefficient (outstrength, $\chi^2 = 13.691$, $P_R = 0.002$; clustering coefficient, $\chi^2 = 38.130$, $P_R < 0.001$; Fig. 2). Specifically, males and females initiated similar amounts of social interactions (outstrength) when on nutritional environments that were low-calorie or protein-rich, but males initiated more social interactions than females on nutritional environments that were high-calorie or carbohydrate-rich (sex-by-caloric concentration interaction, $\chi^2 = 4.591$, $P_R = 0.033$; sex-by-P:C ratio interaction, $\chi^2 = 9.599$, $P_R < 0.001$; Fig. 2). Females increased how cliquish they were (clustering coefficient) in high-calorie and carbohydrate-rich environments, while males were less cliquish in low-calorie environments and maintained similar levels of cliquishness across environments of varying protein-to-carbohydrate ratio (sex-by-caloric concentration interaction, $\chi^2 = 33.250$, $P_R < 0.001$; sex-by-P:C ratio interaction, $\chi^2 = 25.890$, $P_R < 0.001$; Fig. 2). The nutritional environment also played a significant role on its own in affecting instrength, outstrength, clustering coefficient, and variation in eigenvector centrality. Individuals interacted less overall (lower instrength and outstrength) on nutritional environments that were more carbohydrate-rich compared to protein-rich environments (instrength, $\chi^2 = 5.175$, $P = 0.023$; outstrength, $\chi^2 = 8.122$, $P = 0.004$), and individuals tended to be more cliquish (clustering coefficient) overall on high-calorie ($\chi^2 = 11.618$, $P < 0.001$) and carbohydrate-rich environments ($\chi^2 = 7.057$, $P = 0.008$).

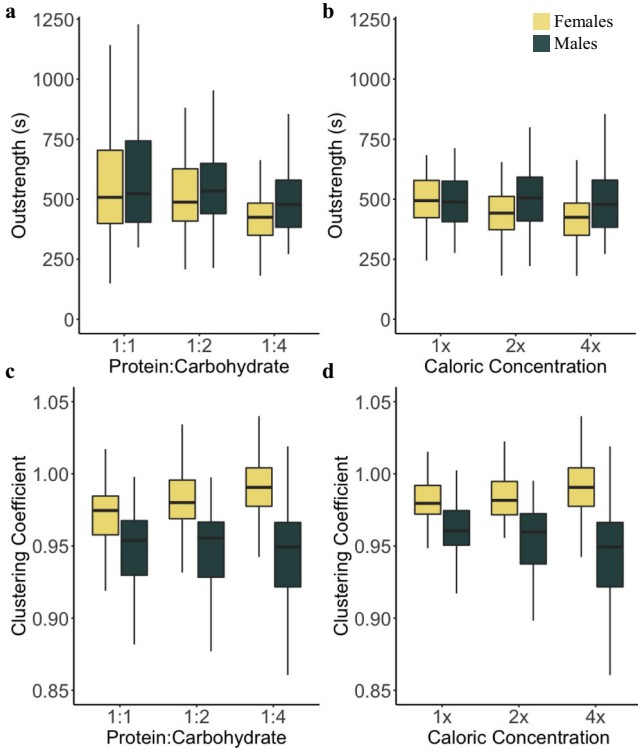

**Fig. 2 Sex-by-nutritional environment effects on network position.** Sex differences in outstrength (**a**, **b**) and clustering coefficient (**c**, **d**) significantly differed across nutritional environments of varying protein-to-carbohydrate ratio (**a**, **c**) and caloric concentration (**b**, **d**). Boxplots present the medians (horizontal lines), interquartile ranges (boxes), and values within ± 1.5x IQR (whiskers) for each sex in each nutritional environment (1:1–1:4, 1x–4x; $n = 160, 120, 130, 50, 100, 130$, respectively for each sex). Sexes are delineated by the color fill of the boxplots (yellow = females, navy = males). Source data are provided as a Source Data file.

In high-calorie environments, we detected increased variation in how central individuals were to the structure of their social groups (Fligner-Killeen test: eigenvector centrality, $\chi^2 = 8.097$, $P = 0.017$). Notably, these patterns of genotypic, sex, genotype-by-environment, and sex-by-environment effects on network position could not be attributed to differences in the overall locomotor activity of each fly (Supplementary Table 2). For a sub-sample of social groups in which network structure was measured across two days, we found that individuals displayed significant consistency across days in four of the five network positions analyzed (Kendall's rank correlations: instrength, $\tau = 0.167$, $z = 4.017$, $P < 0.001$; outstrength, $\tau = 0.222$, $z = 5.337$, $P < 0.001$; clustering coefficient, $\tau = 0.458$, $z = 11.007$, $P < 0.001$; and eigenvector centrality, $\tau = 0.091$, $z = 2.179$, $P = 0.029$) but not for betweenness centrality ($\tau = 0.024$, $z = 0.524$, $P = 0.6$), further supporting our inference that genetic variation contributed to individual differences in network positions.

For a trait to respond to selection and evolve, it must have an additive genetic basis[44]. Our findings that genotypes significantly vary in all measures of social network position indicate network structure can likely evolve in response to selection. Prior studies estimating the heritability of social network positions have reached variable conclusions about which measures of network position are heritable[16–18]. In wild populations, estimates of heritability may be confounded by prior experience[11,34,39,45,46], social group composition[11,14,22,25,45,46], and carryover effects (e.g., maternal effects or social inheritance)[47,48]. Here, we demonstrate that social network positions are heritable under

controlled laboratory conditions, and with direct replication of social groups of natural genotypes. Interestingly, our broad-sense heritability estimates were generally comparable to prior heritability estimates of social network positions in the wild[16–18].

**Fitness and selection on social network positions.** To further probe the evolutionary potential of social networks, we quantified selection gradients for social network position, and how these gradients varied across nutritional environments. For males, we observed directional selection acting on network position measures of instrength and eigenvector centrality. For males, receiving more interactions (instrength) and being more central to group structure (eigenvector centrality) were associated with lower mating success overall (type III Wald $\chi^2$ tests: instrength, $\chi^2 = 8.795$, $P = 0.003$; and eigenvector centrality, $\chi^2 = 9.938$, $P = 0.002$; Supplementary Table 3). Additionally, males who received more interactions (instrength) had longer latencies to mate ($\chi^2 = 9.687$, $P = 0.002$). We also found that the strength and direction of selection, measured as the relationships between male fitness measures and instrength, outstrength, and eigenvector centrality, varied across nutritional environments (Fig. 3, Supplementary Table 3). These findings are particularly exciting, as context-dependent selection can maintain genetic variation when the strength and direction of selection on a phenotype changes depending on the environmental context in which that phenotype is expressed[32,36]. In high-calorie environments, males who received more social interactions (instrength) or were more central to the structure of their social groups (eigenvector centrality) also showed both an increased observed mating success (instrength-by-caloric concentration, $\chi^2 = 7.098$, $P = 0.008$; and eigenvector centrality-by-caloric concentration, $\chi^2 = 9.859$, $P = 0.002$; Fig. 3) and shorter latencies to mate (instrength-by-caloric concentration, $\chi^2 = 10.367$, $P = 0.001$; and eigenvector centrality-by-caloric concentration, $\chi^2 = 10.628$, $P = 0.001$). This pattern was reversed in low-calorie environments, where receiving more interactions and being central to group structure was associated with lower mating success and longer latencies to mate. Males who initiated more social interactions (outstrength) had marginally shorter latencies to mate in high-calorie environments, but tended to have longer latencies to mate in low-calorie environments (outstrength-by-caloric concentration, $\chi^2 = 6.471$, $P = 0.011$; corrected significance threshold = 0.01). Similarly, males who were more central to the structure of their social group had higher mating success in carbohydrate-rich environments, but had lower mating success in protein-rich environments (eigenvector centrality-by-P:C ratio, $\chi^2 = 6.678$, $P < 0.010$; Fig. 3); and males who engaged in more social interactions (instrength and outstrength) had shorter latencies to mate in carbohydrate-rich environments, but had longer latencies to mate in protein-rich environments (instrength-by-P:C ratio, $\chi^2 = 7.317$, $P = 0.007$; outstrength-by-P:C ratio, $\chi^2 = 8.183$, $P = 0.004$). We also detected marginally significant effects of increased social interactions (instrength and outstrength) associated with greater male mating success in carbohydrate-rich environments, with the opposite trend occurring in protein-rich environments (instrength-by-P:C ratio, $\chi^2 = 6.478$, $P = 0.011$; and outstrength-by-P:C ratio, $\chi^2 = 5.733$, $P = 0.017$; corrected significance threshold = 0.01; Fig. 3). Male genotypes also significantly differed in their mating success (likelihood ratio test: LRT = 99.76, $P < 0.001$) and latency to mate (LRT = 87.104, $P < 0.001$). Our findings indicate that higher rates of social interactions and centrality to group structure are beneficial in high-calorie and carbohydrate-rich nutritional environments, but costly in low-calorie and protein-rich environments, suggesting that social groups of flies in high-calorie and carbohydrate-rich environments would evolve to be more

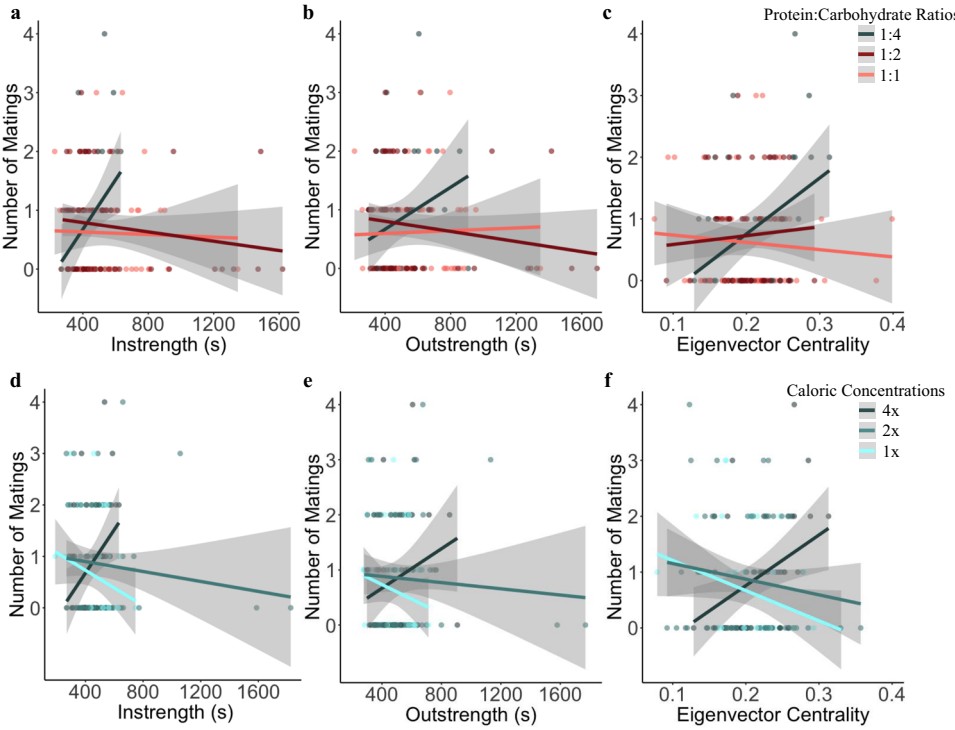

**Fig. 3 Selection on network position is context-dependent.** Selection gradients, quantified as the correlations between male mating success and three network position metrics measured the first day after social groups were established, varied across nutritional environments that varied in protein: carbohydrate ratio (**a**–**c**) and caloric concentration (**d**–**f**). Lines show best fit linear relationships with standard error (shaded areas) (1:4–1:1, 4x–1x; $n = 40$, 60, 60, 40, 70, 50, respectively). The colors of each line correspond to the given protein:carbohydrate ratio or caloric concentration. Source data are provided as a Source Data file.

cohesive and interactive compared to groups in low-calorie and protein-rich environments.

While we found context-dependent and directional selection operating on social network positions in males, we found no effects of network positions on components of female fitness (all $P > 0.05$; Supplementary Table 3). We also observed no differences among female genotypes in lifespan (LRT = 0.16, $P = 0.689$) or lifetime offspring production (LRT = 0.04, $P = 0.842$). However, the nutritional environment social groups interacted on did influence female lifespan and offspring production, with females living longer from low-calorie and protein-rich environments (caloric concentration, $\chi^2 = 10.815$, $P = 0.001$; P:C ratio, $\chi^2 = 7.443$, $P = 0.006$) and producing more offspring after interacting on protein-rich environments (P:C ratio, $\chi^2 = 4.860$, $P = 0.028$). These results are largely consistent with past work showing caloric restriction and short-term stress increases lifespan, and elevated dietary protein increases reproduction[40,49]. However, our findings are particularly noteworthy, as flies were only on their treatment nutritional environments for less than 3 days out of their total lifespan (average female lifespan 60 days ± 27 standard deviation), and were otherwise reared and housed on standard lab food[39].

## Discussion
Our findings of context-dependent and directional selection operating on heritable network positions were evident only for males, and not for females. One potential hypothesis explaining this pattern is that females may be better able to adjust their behavior to adaptively match their environment, compared to males[9,36,46,50,51]. This hypothesis is consistent with our findings of sex-by-environment interactions for network positions. An alternative hypothesis for our sex-by-environment interaction

results could be that males and females perform different social functions depending on the environment, irrespective of the fitness consequences of engaging in different social behaviors. This would align with previous work demonstrating sex-specific patterns of gene expression change in response to social and nutritional cues from the environment[52,53]. While our study utilized a quantitative genetic approach to addressing how genotype influences measures of social network position, this genotype-to-phenotype relationship is likely complex and involves many intermediary links. Understanding the links between genotype and social network position is an active area of study, with recent work exploring everything from how genes for sensory processing affect network structure[54,55], to how individual behavioral differences inform network structure (e.g., exploratory differences, group size preferences, and sociability)[10,56]. Future work should continue to explore the genetic and behavioral underpinnings of network structure, as well as how these effects may vary across environmental contexts.

Across a diverse range of taxa, evidence that social network positions are under selection is mounting (mating success[4,5], disease risk[1,6], effects on foraging[8,9], review[27]) in tandem with evidence for remarkable intra-population variation in these same traits[20,26–31]. How can heritable variation in social network positions be maintained in the face of selection and drift? While we found some support for the hypothesis that genotype-by-environment interactions influence measures of social network position (namely how cliquish individuals were, i.e., clustering coefficient), we did not find evidence of selection acting on variation in cliquishness. However, we did find support for the hypothesis that context-independent genetic variation in network position, coupled with context-dependent selection, can maintain heritable genetic variation in measures of social network positions. Studies estimating how selection varies across space or time

are severely underrepresented for behavioral traits[57], and very few have obtained multiple measures of selection on social network positions[58,59]. The strength of our methodology rests in our ability to create replicate social groups of identical genotypic composition. This allowed us to investigate processes that may be obscured by genetic variation in social group membership in natural populations. Our results, in conjunction with prior estimates of the heritability of social network positions[16–18], indicate that the evolutionary potential of social network structure may be a more widespread phenomenon than currently known. It is important to note that measures of social network position are all derived from the same underlying network of social interactions. Thus, if any single measure of network position is heritable, the entire underlying social network is likely to change in response to selection. Future studies investigating the heritability of social network positions across multiple species, populations, and social groups will hopefully bring us closer to understanding the evolutionary potential of social group structure. Continued work should also seek to address how genetic and environmental factors shape the evolution of complex social behaviors, to develop a better understanding of the behavioral, ecological, and evolutionary factors shaping diversity in social group structure. Our findings suggest that studying how both social and ecological factors interact to shape variation in and selection on social phenotypes is necessary for understanding the evolution of social dynamics.

## Methods

**Study system and social groups.** To create replicate social groups, 40 inbred lines of flies were randomly chosen from the *Drosophila* Genetic Reference Panel (DGRP), a panel of dozens of inbred homozygous genotypes derived from a wild population[38]. These 40 inbred lines were randomly divided between maternal and paternal lines, and 20 mating crosses were randomly generated to produce 20 replicate heterozygous genotypes (maternal lines 136, 208, 237, 306, 315, 335, 360, 387, 786, 852, 153, 235, 307, 378, 57, 59, 637, 732, 801, and 855; crossed with paternal lines 427, 313, 714, 774, 229, 365, 380, 324, 716, 721, 517, 712, 176, 765, 862, 820, 318, 40, 513, and 391; respectively). Note that parental line numbers are arbitrary and unindicative of relationships between genotypic lines. Using replicate heterozygous genotypes allows us to alleviate the potentially deleterious effects of homozygous recessive alleles in inbred lines, and generate individuals representative of a subset of naturally segregating genetic variation found in the wild[60,61]. This approach thus balanced our need to create replicated social groups against the challenges of adequately representing natural genetic variation with inbred lines, which may or may not be perfectly representative of quantitative genetic parameters in wild populations. Ten males and ten virgin females of each inbred line were paired and placed in vials of standard fly food to control for maternal and larval density, and heterozygous offspring were allowed to develop for 2-3 weeks. Newly emerged virgin females and males were collected under light $CO_2$ anesthesia from the 20 mating crosses (10 replicate genotypes per sex). Each fly from each sex was randomly marked with a unique color identifier on its mesothoracic segment to visually identify each individual. Flies were aged in same-sex groups for three days to allow for development to sexual maturity and recovery from $CO_2$ anesthesia[62,63]. All flies were reared and aged on a 12:12 light:dark cycle, at 24 °C and 50% relative humidity, and on standard fly food. On the evening of the third day, the 10 males and 10 females were anesthetized via chilling and combined into a social group. Care and treatment of all flies complied with all relevant ethical regulations.

**Nutritional environments.** Each social group was placed in a 10 cm petri dish filled with one of five nutritional environments that varied in two dimensions of nutrient composition: protein-to-carbohydrate ratio (P:C ratio) and caloric concentration[64,65]. Three nutritional environments were constant in the total amount of calories they contained, but varied in whether the calories were derived from protein or carbohydrates (1:1, 1:2, and 1:4 P:C ratio). To vary caloric concentration, three nutritional environments contained a constant ratio of P:C (1:4), but varied in caloric concentration by 4x, 2x, and 1x. Note that the 1:4 P:C ratio and the 4x caloric concentration environments are the same. All recipes contained a base of 27 g agar, 11.1 mL tegocept acid mix (70 g tegocept/270 mL $H_2O$), and 3 mL propionic acid; per 1 L $H_2O$. Caloric concentration and P:C ratio were manipulated by adjusting the amounts and ratio of nutritional yeast and malt sugar added to the foods (1:1 P:C, 4x Concentration = 146.9 g yeast and 45.6 g sugar; 1:2 P:C, 4x Concentration = 97.9 g yeast and 94.6 g sugar; 1:4 P:C, 4x Concentration = 58.8 g yeast and 133.7 g sugar; 1:4 P:C, 2x Concentration = 29.4 g yeast and 66.9 g sugar; and 1:4 P:C, 1x Concentration = 14.7 g yeast and 33.4 g sugar)[40,41].

**Fly behavior and social network analysis.** Social groups were given a period of overnight acclimation, then video-recorded for 20 min during the hour immediately following lights-on (when flies are most active) over the course of two days[66]. Videos were recorded with Nikon D3300 cameras at 30 fps and 9.97 ± 0.50 pixels/mm. Each video was processed with the motion-tracking software Caltech Fly-Tracker 1.0.5, which outputs the position and orientation of every fly in every frame of a video[42]. We manually verified the tracking output for all individuals to ensure all tracking identities were consistent and accurate. We also manually validated the tracking output of the software by comparing hand-annotated fly positions to FlyTracker's output in a stratified random subsample of 700 frames (correlation > 0.999).

Using the tracking output from FlyTracker, we calculated the weighted and directed social interactions occurring between every pairwise combination of flies to build a social interaction matrix for each video[67]. A focal fly was considered interacting with another fly if three criteria were met: (1) the distance between the two flies was <2.5 average fly body lengths, (2) the interacting fly was within a 320° field-of-view of the focal fly, and (3) if the aforementioned criteria were met for a minimum duration of 0.6 s[68]. Note that our first criterion ensured that the radius in which flies were considered to be interacting was fixed for all individuals. For example, larger flies (i.e., females) were not assigned a larger space in which interactions were counted, which could bias our results. Our second criterion allows for one fly to be in another's field-of-view, without reciprocally having the other fly in its field-of-view; thus, we can distinguish between the directedness of social interactions occurring. While proximity between individuals does not necessarily imply a social interaction occurred, these interaction criteria filter out random interactions, such as two flies walking quickly past one another without socially engaging[68]. Edges in our interaction matrices were defined as the total duration of time all pairwise combinations of flies spent interacting throughout the duration of a video. Since all individuals were observed at all times throughout a video, the interaction networks we built allowed us to directly proceed with network analysis without having to compute association indices to account for sampling error or bias[69,70]. We chose to analyze the five most commonly studied individual-level social network positions using the R package igraph 1.2.4.2[71]: instrength and outstrength—the amount of time other individuals spend socially engaging with a focal individual, and the amount of time a focal individual spends socially engaging with other individuals, respectively; weighted and directed clustering coefficient—how interconnected a focal individual's direct social partners are to one another (i.e., cliquishness); weighted and directed betweenness centrality—the number of shortest paths between any two individuals that transverse a focal individual; and weighted and directed eigenvector centrality—how critical a focal individual is to the overall structure of the group based on the strength of its direct social connections, the strength of its partners' connections, the strength of its partners' partners' connections, etc.[1–3].

**Measures of fitness.** We measured two components of fitness for males and two components of fitness for females, to understand how selection acts on social network phenotypes across different environmental contexts. In the two hours immediately after males and females were combined into a social group, the identities of all mating pairs and their latency to mate were recorded. The total number of matings and latency to first copulation constitute the two measured components of males' fitness. Females were rarely observed to remate (2/507 observed copulations). On the morning of the third day after social groups were established, females were removed from their groups using cold anesthesia and isolated in individual vials containing standard fly food. Females were transferred to new vials with fresh food medium every week until death, and lifespan was recorded as our first component of female fitness. After a female was removed from a vial, we allowed eggs and larvae to develop and counted all eclosed adult offspring up to 21 days after a female was first introduced to a vial. The minimum duration from egg to adult in *D. melanogaster* is approximately 11 days[72], so we were able to ensure all counted offspring were the female's progeny and not F2 individuals. By transferring females to a new vial every week before any offspring eclosed, we were able to ensure that all eggs produced were derived from matings that occurred when the female was in the social group. The total lifetime offspring production of each female constituted our second component of female fitness.

**Replication.** Ninety-eight replicate social groups were created and divided amongst the five nutritional environment treatments (1:1 P:C/ 4x Concentration, n = 24; 1:2 P:C/ 4x Concentration, n = 18; 1:4 P:C/ 4x Concentration, n = 14; 1:4 P:C/ 2x Concentration, n = 22; 1:4 P:C/ 1x Concentration, n = 20). If any flies died or escaped, the social group they were a part of was excluded from being videoed, resulting in the exclusion of almost half of our replicates from network analysis. This was necessary however, as uncontrolled dynamics within the group (e.g., presence of a dead individual affecting living social group members) likely contribute variation to measures of social group structure and cause groups to no longer be truly replicated. For intact social groups, up to two videos were taken (one/day over two days). Forty-three independent social groups had fully tracked videos, 13 of which were videoed on both days, giving us 56 fully tracked videos of social groups to be used for network analysis (1:1 P:C/ 4x Concentration, n = 16; 1:2 P:C/ 4x Concentration, n = 12; 1:4 P:C/ 4x Concentration, n = 13; 1:4 P:C/ 2x Concentration, n = 10; 1:4 P:C/ 1x Concentration, n = 5). All flies from a group

were excluded from fitness analyses if any flies in their group died or escaped before we began to quantify their fitness metrics. For analyses of total number of matings and latency to first copulation, 640 males were used, including males that were never observed to mate (number of matings = 0, latency to mate was right-censored at 120 min). Of these, 340 males also had measured network positions. For analyses of female lifespan and lifetime offspring production, 354 females were used, excluding females that either died during cold anesthesia while being removed from their social groups or escaped after they were removed from their social groups and into individual vials. Of these, 247 females also had measured network positions.

**Analyses of social network positions**. All analyses were conducted in R version 3.6.2[73]. For social network positions of instrength, outstrength, and betweenness centrality, we ran Poisson-distributed generalized linear mixed models (GLMMs), as these network positions are counts. For clustering coefficient and eigenvector centrality, we ran linear mixed models (LMMs)[74–76]. All LMMs and GLMMs were constructed using R package lme4 1.1[77]. In models for instrength, outstrength, betweenness centrality, and clustering coefficient, fixed effects of sex, nutritional environment, and sex-by-environment interactions; and random effects of genotype and social group ID were included. For models of eigenvector centrality, we included fixed effects of sex and sex-by-environment interactions; and random effects of genotype and individual ID to account for non-independence of multiple measures of the same individuals. Group-level properties (e.g., nutritional environment and social group identity) are not able to meaningfully influence mean measures of eigenvector centrality, as this is inherently a relative measure. Model fits were assessed using R package DHARMa 0.2.7. Accommodations for zero-inflation were applied to models of betweenness centrality using R package glmmTMB 1.0.0[78], and accommodations for overdispersion were applied to models for instrength and outstrength using an observation-level random effect[76]. Fixed effect interactions were assessed using Type III Wald $\chi^2$ tests, and non-significant interactions were removed from our models[74]. The resulting models for social network positions constitute our base models, from which further investigations are based. To test for effects of genotype and genotype-by-environment interactions on social network positions, the base models were compared to models excluding genotype and including a genotype-by-environment random slopes interaction, respectively, using likelihood ratio tests[74]. To test for fixed effects of nutritional environment and sex on social network positions, Type III Wald $\chi^2$ tests were employed[74]. Since group-level properties such as nutritional environment cannot influence mean levels of eigenvector centrality, but can influence overall variation, Fligner-Killeen tests were used to test homogeneity of variance in eigenvector centrality across the nutritional environments[79]. Since network data is non-independent, significance of within-group effects (sex, genotype, sex-by-environment interactions, and genotype-by-environment interactions) on social network positions was tested by comparing the observed test statistics to test statistics from 1000 null networks using a one-tailed $t$ test (significance values reported as $P_R$ in text; Supplementary Table 1)[80,81]. Null networks were generated by permuting sex within social groups, for assessing the significance of sex and sex-by-environment interactions; and permuting genotype within each sex within social groups, for assessing the significance of genotype and genotype-by-environment interactions. To better understand the behavioral processes underlying variation in social network positions, we tested for effects of sex, genotype, nutritional environment, sex-by-environment interactions, and genotype-by-environment interactions on fly locomotor activity (measured as the total distance a fly moved in a video) using LMMs with fixed effects of nutritional environment, sex, and their interaction; and random effects of genotype and social group ID. Upon finding significant effects of sex ($\chi^2 = 15.679$, $P_R = 0.001$), sex-by-environment interactions ($\chi^2 = 12.615$, $P_R = 0.003$), and genotype (likelihood ratio = 118.87, $P_R < 0.001$) on activity, we added an activity covariate to our base models for social network positions to clarify that observed patterns are due to social processes and not sex and genotypic differences in activity (Supplementary Table 2). In cases where groups had measures of network structure taken both one day and two days after social groups were established, we tested for individual consistency in network positions using Kendall's rank correlations. Kendall's rank correlations were used since our measures of network positions are non-normally distributed. As our five nutritional environment treatments differed in two dimensions of nutrition (protein-to-carbohydrate ratio and caloric concentration), we sought to further understand how variation in nutritional environments affects social network positions. We analyzed the effects of P:C ratio and caloric concentration in separate models using the previously described base models for individual-level network positions, only with P:C ratio and caloric concentration levels substituted for the effect of the nutritional environment. Broad-sense heritability ($H^2$) estimates for social network positions were acquired by estimating the proportion of total variance explained by the random effect of genotype in our base models using R package MuMIn 1.43.17[82–84]. Current methods of estimating the proportion of variance explained by random effects are unequipped to handle zero-inflated models, so our base model for betweenness centrality was amended with an observation-level random effect to account for the excess variation attributed to zero-inflation.

**Fitness analyses**. For number of matings, lifespan, and lifetime offspring production, we ran Poisson-distributed GLMMs, as these fitness components were either counts or best represented by a Poisson distribution[74,76]. For latency to first copulation, we ran mixed effect Cox proportional hazards models using the R package coxme 2.2, as our latency data was right-censored[85]; nearly half of males (331/640, 48.5%) did not mate in our initial two hours of observations, meaning their latency to mate was an unknown duration of >2 h. We confirmed that our data met the assumptions of Cox proportional hazards models by fitting a survival function including a variable indicating whether each measurement was censored or not (all global $P > 0.05$). Initial models for each fitness component contained a fixed effect of nutritional environment, and random effects of genotype and social group ID. Model fit for GLMMs was assessed using R package DHARMa 0.2.7. Accommodations for zero-inflation were applied to models for number of matings, and accommodations for overdispersion were applied to models for lifetime offspring production and lifespan by specifying a negative binomial distribution using R package glmmadmb 0.8.3.3[86]. These models constitute our base fitness component models from which further analyses are based. We used likelihood ratio tests to test the effect of genotype, and Type III Wald $\chi^2$ tests to test the effect of nutritional environment in each of our base fitness component models. For models testing the effects of each social network position and its interaction with the nutritional environment on our fitness components, we only considered individuals that had both a fitness response and measured network phenotypes. Each individual fly had only one possible measurement of fitness per fitness component, but individuals could have two measures of each social network position (in cases where videos of the social group were taken both one and two days after social groups were formed). Because individuals displayed significant consistency in network positions across days, we used network data collected on the first day after social group formation in our fitness analyses, as the first day of network data had a more robust sample size (Supplementary Table 4). Social network positions are often highly correlated, which can create problems when multiple collinear variables are included in multiple regression models[87]. We tested for Kendall's rank correlations between all pairwise combinations of our five social network positions, and found all to be significantly correlated (Supplementary Fig. 2). To test if the observed multicollinearity would pose problems in multiple regression models, we measured the variance inflation factors of the social network positions by adding all network positions to the base models for each fitness component. The observed multicollinearity was far above amounts generally considered acceptable for multiple regression analyses[87]. As such, we analyzed the effect of each of the five social network positions on each fitness component in a model on its own, and applied a Bonferroni correction for multiple testing to $P$ values from these models. Because our measures of network position are highly multicollinear and non-independent, significance tests will likely be conservative with corrections for multiple testing[88]. Each of the social network positions and its interaction with the nutritional environment were added to the base models for each fitness component. If the two-way network position-by-environment interaction effect was found to not affect fitness, the interaction was removed from the model. The significance of each social network position and network position-by-environment interaction was assessed using Type III Wald $\chi^2$ tests[74]. To better understand how variation in nutritional environments affect the strength and direction of selection on social network positions, we analyzed the effects of our two dimensions of nutrition separately by substituting the effects of P:C ratio and caloric concentration for nutritional environment in the models described above (Supplementary Table 3).

**Reporting summary**. Further information on research design is available in the Nature Research Reporting Summary linked to this article.

## Data availability
Source data for all figures and tables are provided with this paper. All other data is publicly available on Zenodo repository (https://doi.org/10.5281/zenodo.4642991)[89]. Source data are provided with this paper.

## Code availability
Source code for this work is publicly available on Zenodo repository (https://doi.org/10.5281/zenodo.4642991)[89].

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

## Acknowledgements
We thank V.H.W. Rudolf, S.P. Egan, F. Oswald and the Saltz laboratory for discussions; E. Dullea, S. Maddox, N. Koonce, Q. Tran, J. Beshai, A. Geiger, and R. Anderson, for data collection assistance; N. Pinter-Wollman, T.W. Wey, and D.N. Fisher for assistance with network and statistical analysis; and SJ Art and Frames for donating supplies. This research was supported by the National Science Foundation (IOS-1856577 to J.B.S.), a Society for the Study of Evolution Rosemary Grant award to E.W.W., and a Houston Livestock Show and Rodeo award to E.W.W.

## Author contributions
E.W.W. designed, performed, and analyzed most of the research. J.B.S. assisted with experimental design and analysis, and supervised the research. E.W.W. wrote the manuscript with assistance from J.B.S.

## Competing interests
The authors declare no competing interests.
