## [Peer Review File · Nature Communications]

REVIEWER COMMENTS

Reviewer #1 (Remarks to the Author):

I would like to begin my review with a bit of historical context. A few years ago, when I submitted a similar paper on the heritability of social network position to Nature Communications the editorial team rejected it because of the claims of an editor involved in the decision (not reviewer, *editor* of this journal) that heritability of any trait will track toward zero as the sample size of the study increases. They also later published a flawed paper (#46 in the reference list of the current paper) that uses an analysis that does not allow for genetic effects and then concludes that only maternal effects are relevant in social systems. That paper is a dangerous throw-back to 'nature v. nurture' and it is a puzzle why it was published. On balance of past experience, this journal therefore appears to be 'team nurture' when it comes to social behavior (as though it were 50 years ago!). I therefore put forward the message to these authors to move on swiftly should the editorial decision not go their way.

Now on to my thoughts on this paper (and for what they might be worth to this journal, particularly after I've written the above!). This is a well written piece based on data impressive in their scope and depth. The stage is set very nicely with regards to the problem at hand and why it is difficult to answer in non-laboratory systems. The result that nutritional environment did not significantly influence the network position of individuals of different genotypes, but did influence network position of individuals in general, is particularly interesting and important. So too are the fitness results, whereby the social network position of males has a differential relationship to fitness outcomes in different environments. This result uncovers long postulated ideas about context-dependency maintaining variation in everything from social network position, personality, to mood-related disorders that influence individual variation in social behaviour in humans. In short, this is an excellent paper that I would like to see published. I have just a few questions that I would like clarified and a few suggestions for improvements to this piece.

The methods state that the 5 network measures used here are highly correlated, to the level that this raised issues in a VIF model. To deal with this in the fitness analysis, a Bonferroni correction was applied. But does this collinearity not also impact the rest of the results and their interpretation? That is, does it not also cause issues for statements made about the heritability of these measures, and the implication that these are independent results, and thus each distinct and important in their own right?

I struggled to understand what in-strength and out-strength represent in this system. These directional network metrics are meant to represent the giver or initiator of behaviours (out-strength) vs the receiver of behaviours (in-strength). However, it wasn't clear from the description of behaviours given how giving/receiving were defined or recorded (or indeed if they are even relevant for the behaviours observed). More information on this is needed.

It would be helpful to have the link (or lack thereof) from genotype to network position to context spelled out in clearer terms as this is an important part of the results.

Similarly, I would suggest adding a line or two to further unpack the findings that nutritional environment did not significantly influence the network position of individuals of different genotypes, but did influence network position of individuals in general. This is important and therefore it is important that readers fully grasp how this might come about and what it might mean.

Finally, the discussion needs to return to the methodological aspects of this study that sets it apart from past studies on non-captive populations. That is, the discussion should return to mentioning the fact that the networks in this study were composed of genetically indistinct individuals and should highlight what that means for the likely generalisability of these results in more natural settings, and therefore their implications.

Reviewer #2 (Remarks to the Author):

This study addresses the very interesting and important topic of whether social network position is a heritable trait, whether selection acts on this trait, and whether there are interactions with the environment. The study makes use of an experimental design where replicated groups of fruit flies containing the same 20 genotypes were created in the lab. These groups were recorded and their social network structures measured, as well as their fitness outcomes including eggs produced (females) and matings (males). The results indicate a significant effect of the random effect “genotype” in the models, which is interpreted as evidence of heritability of individual network positions, and this effect was consistent across environments (no GxE effect). There were also significant effects of sex on network position. In regard to the fitness measurements, the study reports directional selection on network position for males as well as context-dependent selection depending on the environment. These effects were not found in females.

Overall, the study is highly innovative and tackles an important topic, and the experimental methods of replicated social groups are a powerful approach to get at questions of heritability of social behavior. The article is also well-written and clear. I do have a few critiques and comments, however.

First, I found that the study suffered a bit from trying to do too much, and unfortunately this somewhat diminished the power of the replicated social groups. In particular, a simpler experimental design (fewer conditions) allowing simpler statistics might have allowed the hypothesis of heritability of network position to be tested in a more robust way, rather than having to make use of the complex regression models that typically need to be used in ecological settings when fewer things can be controlled. This

also relates to my statistical comments below.

Secondly, although the study purports to be about network position, it is not clear to me that this is clearly differentiable from overall “sociability” (or anyway, proximity) of individuals. For instance, in strength and out strength both capture the overall level of sociability of an individual, and the other network measures are highly correlated with these metrics. I realize this is a challenging topic to address, but to me the emphasis on network position somewhat sweeps under the rug the possibility that the patterns might be driven by something simpler, e.g. overall sociability of individuals. I also wonder whether other unmeasured factors, such as spatial positioning in the experimental arena, could be driving these patterns. To its credit, the study does test for the effect of overall activity level. However, I think these potentially confounding factors and subtleties about network position vs sociability should be mentioned in the discussion. In addition, while proximity is often used as a proxy for social interaction in animal social networks studies, it does not necessarily imply it and this limitation should also be discussed.

Finally, I have some concerns about the statistics. Overall, a large number of statistical tests have been conducted, increasing the possibility of false positive results, which does give me some overall cause for concern. For instance, the use of both day 1 and day 2 networks as separate predictors in the models seems not fully justified.

On a somewhat related note: are these two days correlated? It would make sense that, if social network positions are a real phenomenon here, that these would be somewhat repeatable across days, and this would be worth investigating.

Secondly, I have some concerns about the permutation tests described in the methods. It would be nice to have a bit more explanation of what exactly was done here. From my understanding, the authors assessed significance of all the effects by comparing the test statistic associated with each metric to the distribution of test statistics for that metric in models where individual identity labels had been permuted within groups (?). However, it would seem that this randomisation might not make sense for all comparisons. For instance, for looking at the effect of the random factor genotype while controlling for sex, a reasonable randomisation would be to permute the labels within each sex separately (rather than mixing them). Also, I wondered about the possible effects of individual consistency (i.e. consistency not within a genotype but within a single individual in network position) that could be confounded here, since individual ID is (understandably) not included as a random effect but network measurements for each individual have been replicated twice (therefore genotype ID might pick up some of the variation that actually should be attributed to individual ID here).

To address some of these concerns, I suggest that an additional, simpler analysis addressing the main question of whether social network position is heritable be conducted with the existing data, as a sanity check. This test would make use of the fact that social groups have been completely replicated multiple times in the same conditions (a powerful feature of this study!), and simply check whether the network metrics associated with different genotypes are repeatable WITHIN each experimental condition,

controlling for sex. In other words, can you show directly that when you don't change the environment, individuals of the same genotype are more similar in their network positions than individuals of different genotypes? This could be done by computing some test statistic associated with a given metric (e.g. the sum of the variation in instrength across individuals of the same genotype, within a particular environmental condition), then permuting identities within each social group amongst individuals of the same sex to generate a null distribution of this test statistic. This would avoid the need for complicated regression models to address that central question, and in my view would strengthen the results and better make use of the replicated social groups.

Additional comments:

Figure 1 & 2: The wording of what the box plots show is a little unclear - what is meant by +/- 1.5x IQR? Also, how far do the lines extend - presumably 95% range?

L411-413: More clear definitions of what was extracted from the data are needed here. What was the edge definition? Or, in the case of instrength and outstrength, what is the metric computed - fraction of time being interacted with / interacting with others? Also, definitions of the other 3 network metrics should be given, since these can be computed slightly different ways.

L449: Would this not induce a bias in the data, since males that never mated are not included in the latency analysis? Perhaps one could use survival analysis with right-censored data to get around this issue?

Methods - Replication: There is quite a lot of missing data in this study, and a casual reader might not notice this. Could this be mentioned in the results and/or discussion, as a possible limitation? The information in this section could also be summarised in a table, which might aid the reader.

L442-443: Since there were only 56 fully tracked videos in the end (out of a possible $98 \times 2 = 196$), how many of these were independent groups (vs replicates of the same group)?

Reviewer #3 (Remarks to the Author):

In this very interesting paper, the Authors conduct a genetically controlled assay on heritability and effect of environmental factors (nutrients available) of social network features. They show that multiple measures of network position are inheritable and that sex interacts with the environment in determining the relation between fitness and social network measures.

I enjoyed reading the paper. There are a few points that I believe should be clarified and addressed.

Would it be worth to quantify to which extent genotype explains variability in social network position?

the fact that one crossing and F1 is representative of a natural population should be justified/discussed
69: related to the point above: they are not really identical

13 and in the manuscript: you refer to multiple measures of social networks... this is a bit vague. Can you focus on those that are independent from each other?

134-138, 142-143: these experiments have been conducted on very different species, it might be a misleading comparison.

Fig 3: datapoints should be visualised

225: aren't social pressures a kind of selective pressure? it seems you argue that natural selection might work using different principles

extended data figure 2:

- I wonder whether readability can be improved for the scatterplots

Minor

5-6 why not mentioning the traits investigated in this paper?

19-22 can this part be more clear on your findings? (e.g. referring to 218-219)

24: what do you mean with "confounds"?

Dear Colleagues,

Thank you very much for your insightful and helpful comments on our manuscript,
“Selection on heritable social network positions is context-dependent in *Drosophila*
*melanogaster*.” In response to your comments, we have revised many sections of the
manuscript, and we believe your comments and our revisions have substantially
improved the manuscript. In particular, we:

- • Amended our permutation tests to randomize genotype within sex within social
groups
 - ○ In doing so, we now find an effect of genotype-by-environment interactions
 - for one measure of network position, and discuss this more thoroughly. All
 - other results are qualitatively unchanged.
- • Revised our analyses of male mating latency to include males who were never
observed to mate, using Cox proportional hazards models with right censoring
 - ○ We now find that our results for how network positions influence male
 - mating latency is more in line with our results showing how measures of
 - network position influence males’ mating success.
- • Added an analysis of individual consistency in network positions across days
 - ○ We find network positions to be largely consistent across days
- • Better visualized data points in Fig. 3 and Supplementary Fig. 2
- • Included a new Supplementary Table 4 to outline our sample sizes of networks
- across days and nutritional environments
- • Addressed how network position multicollinearity can inform our interpretations of
- how network structure evolves
- • Discussed the genotype-to-phenotype relationship for social network positions,
- specifically how individual differences in behavior can inform network positions
- • Grounded/differentiated our results with/against prior studies
- • Clarified wording in multiple sections and figures

We believe these changes have substantially strengthened the findings of this
manuscript, and greatly improved its readability for a broad audience. Additionally,
we’ve taken especial care to revise the manuscript to meet the formatting requirements
of *Nature Communications*. This mostly involved stream-lining the abstract, and
amending the overall structure of different sections.

See below our point-by-point response to reviewers’ comments. Our responses are
italicized and written below each reviewer point. We’ve included a revised version of the
manuscript with changes indicated by the Track Changes function in Microsoft Word.
Thank you for your time spent reviewing our work. We hope that, with any additional
changes you suggest, our manuscript may eventually be published in *Nature*
*Communications*.

Thank you,

Eric Wice

Corresponding author

eric.wesley.wice@gmail.com

**REVIEWER COMMENTS**

Reviewer #1 (Remarks to the Author):

I would like to begin my review with a bit of historical context. A few years ago, when I
submitted a similar paper on the heritability of social network position to Nature
Communications the editorial team rejected it because of the claims of an editor
involved in the decision (not reviewer, *editor* of this journal) that heritability of any trait
will track toward zero as the sample size of the study increases. They also later
published a flawed paper (#46 in the reference list of the current paper) that uses an
analysis that does not allow for genetic effects and then concludes that only maternal
effects are relevant in social systems. That paper is a dangerous throw-back to 'nature
v. nurture' and it is a puzzle why it was published. On balance of past experience, this
journal therefore appears to be 'team nurture' when it comes to social behavior (as
though it were 50 years ago!). I therefore put forward the message to these authors to
move on swiftly should the editorial decision not go their way.

*Thank you for the feedback! We hope our manuscript exemplifies how social behavior*
*isn't a 'nature vs. nurture' issue, and is a product of both processes and their complex*
*interactions. We've also incorporated a new reference of empirical data (Wooddell et al*
*2020 on line 192) to support our assertion that maternal effects and social inheritance*
*can contribute to variation in social network positions (but is certainly not the only*
*process contributing variation).*

Now on to my thoughts on this paper (and for what they might be worth to this journal,
particularly after I've written the above!). This is a well written piece based on data
impressive in their scope and depth. The stage is set very nicely with regards to the
problem at hand and why it is difficult to answer in non-laboratory systems. The result
that nutritional environment did not significantly influence the network position of
individuals of different genotypes, but did influence network position of individuals in
general, is particularly interesting and important. So too are the fitness results, whereby
the social network position of males has a differential relationship to fitness outcomes in
different environments. This result uncovers long postulated ideas about context-
dependency maintaining variation in everything from social network position,
personality, to mood-related disorders that influence individual variation in social
behaviour in humans. In short, this is an excellent paper that I would like to see
published. I have just a few questions that I would like clarified and a few suggestions
for improvements to this piece.

*Thank you very much! We're happy to hear you appreciated the "scope and depth" of*
*the manuscript.*

The methods state that the 5 network measures used here are highly correlated, to the
level that this raised issues in a VIF model. To deal with this in the fitness analysis, a
Bonferroni correction was applied. But does this collinearity not also impact the rest of
the results and their interpretation? That is, does it not also cause issues for statements

made about the heritability of these measures, and the implication that these are
independent results, and thus each distinct and important in their own right?

*We'd first like to note that our application of a Bonferroni correction in our fitness*
*analyses likely makes our interpretations of these models hyper-conservative. This is*
*because our network position measures are correlated and non-independent, while*
*Bonferroni corrections are meant for multiple independent tests. While many of our*
*network position measures are colinear (see Supplementary Fig. 2), there is still*
*important and distinct variation between them. This is why we included a correction for*
*multiple testing, because there is important variation between the network position*
*measures that isn't explained by their multicollinearity.*

*To your point about how the multicollinearity affects interpretation of the network*
*models, we think this is extremely interesting from the perspective of trait correlations*
*and the evolution of correlated traits. As network position measures are all derived from*
*the same underlying network of social interactions, network position measures are*
*inherently interdependent and often correlated. Heritability and selection on one network*
*position measure will likely change the structure of the underlying network, thus*
*resulting in changes in other network position measures as well. We think this is an*
*excellent point, as it speaks to a broader understanding of the evolution of social*
*structure. We've added the statements to this effect on lines 321-335.*

I struggled to understand what in-strength and out-strength represent in this system.
These directional network metrics are meant to represent the giver or initiator of
behaviours (out-strength) vs the receiver of behaviours (in-strength). However, it wasn't
clear from the description of behaviours given how giving/receiving were defined or
recorded (or indeed if they are even relevant for the behaviours observed). More
information on this is needed.

*A second reviewer also had trouble understanding how we differentiated between*
*instrength and outstrength, so we apologize for this initial lack of clarity. A focal fly was*
*considered to be initiating an interaction with another fly if three criteria were met (lines*
*395-398):*

- 1. *the distance between the two flies was < 2.5 average fly body lengths*
- 2. *the focal fly had the other fly within a 320° field-of-view*
- 3. *criteria #1 and #2 were met for a minimum duration of 0.6 seconds*

*Because the angle criteria (#2) allows for one fly to be in another's 'field-of-view' without*
*being reciprocally in that other fly's 'field-of-view', we can distinguish between the*
*initiator and receiver of social interactions. We've clarified this immediately following our*
*criteria for what defines a social interaction (lines 398-401). To your point about the*
*relevance of the directedness of interactions to the behaviours observed: while it is true*
*that most social interactions were reciprocal (instrength and outstrength were the most*
*highly correlated of the network measures; see Supplementary Fig. 2), variation in the*
*directedness in social interactions was still of notable importance. For example, while*
*instrength and outstrength were highly correlated, there was still important variation*
*between these measures explained by the sex of the flies (Fig. 1).*

It would be helpful to have the link (or lack thereof) from genotype to network position to
context spelled out in clearer terms as this is an important part of the results.

*The genotype-to-phenotype relationship is complex, and many different links between*
*genotype and social network position are likely at play (e.g. genotype-to-gene*
*expression, gene expression-to-protein expression, protein expression-to-an individual's*
*actions, an individual's actions-to-social behaviors, social behaviors-to-social network*
*positions, etc.). Thus, understanding the intermediary mechanisms between genotype*
*and social network position is outside the scope of this manuscript, as our goal was to*
*establish the evolvability of network structure and how exogenous factors (i.e. the*
*nutritional environment) contribute to shaping the evolution of social group structure.*
*Studying the genotype-to-phenotype relationship, specifically for social phenotypes, is a*
*very active and important area of study though. We've drawn attention to the various*
*ways that genotypic differences can manifest into differences in social network positions*
*by adding statements to this effect on lines 296-302.*

Similarly, I would suggest adding a line or two to further unpack the findings that
nutritional environment did not significantly influence the network position of individuals
of different genotypes, but did influence network position of individuals in general. This
is important and therefore it is important that readers fully grasp how this might come
about and what it might mean.

*Thank you for this suggestion. We've amended and expanded our explanation of GxE*
*for network positions on lines 136-149 to better explain what GxE for network positions*
*means, and we've strengthened our explanation for why GxE is an important*
*component of maintaining genetic variation. Additionally, per another reviewer's*
*suggestion, we amended our permutation tests to assess the significance of genotype-*
*by-nutritional environment interactions (see response to reviewers line numbers 300-*
*328). In doing so, we now find a genotype-by-nutritional environment interaction for the*
*network position of clustering coefficient (but no other measures of network position).*
*This is particularly exciting, and we hope the statements we've added per your*
*suggestions allow the reader to more fully grasp the importance of GxE for measures of*
*network position.*

Finally, the discussion needs to return to the methodological aspects of this study that
sets it apart from past studies on non-captive populations. That is, the discussion should
return to mentioning the fact that the networks in this study were composed of
genetically indistinct individuals and should highlight what that means for the likely
generalisability of these results in more natural settings, and therefore their
implications.

*Thank you for this suggestion. We've incorporated your suggestion by amending our*
*discussion on lines 316-321.*

Reviewer #2 (Remarks to the Author):

This study addresses the very interesting and important topic of whether social network
position is a heritable trait, whether selection acts on this trait, and whether there are
interactions with the environment. The study makes use of an experimental design
where replicated groups of fruit flies containing the same 20 genotypes were created in
the lab. These groups were recorded and their social network structures measured, as
well as their fitness outcomes including eggs produced (females) and matings (males).
The results indicate a significant effect of the random effect “genotype” in the models,
which is interpreted as evidence of heritability of individual network positions, and this
effect was consistent across environments (no GxE effect). There were also significant
effects of sex on network position. In regard to the fitness measurements, the study
reports directional selection on network position for males as well as context-dependent
selection depending on the environment. These effects were not found in females.

Overall, the study is highly innovative and tackles an important topic, and the
experimental methods of replicated social groups are a powerful approach to get at
questions of heritability of social behavior. The article is also well-written and clear. I do
have a few critiques and comments, however.

*Thank you very much for your feedback! We're happy to hear you found our*
*experiments to be an innovative and powerful approach to tackling this important topic.*

First, I found that the study suffered a bit from trying to do too much, and unfortunately
this somewhat diminished the power of the replicated social groups. In particular, a
simpler experimental design (fewer conditions) allowing simpler statistics might have
allowed the hypothesis of heritability of network position to be tested in a more robust
way, rather than having to make use of the complex regression models that typically
need to be used in ecological settings when fewer things can be controlled. This also
relates to my statistical comments below.

*Thank you for this perspective. While multivariate and hierarchical regression models*
*are commonly used in ecological settings with less control, we believe our ability to*
*apply these models to a study with far greater control of system parameters is one of*
*the biggest strengths of our system and experiments. The power of our genotypically*
*replicated social groups, combined with our ability to manipulate and control many*
*conditions without the confounding effects of prior experience, allows us to address*
*multiple questions relating to the evolution of social group structure that would be far*
*more difficult to test in wild study systems. We also believe our experimental design with*
*multiple conditions makes our work more complementary to studies in wild/ecological*
*settings. Our responses to statistical comments are below.*

Secondly, although the study purports to be about network position, it is not clear to me
that this is clearly differentiable from overall “sociability” (or anyway, proximity) of
individuals. For instance, in-strength and out-strength both capture the overall level of
sociability of an individual, and the other network measures are highly correlated with

these metrics. I realize this is a challenging topic to address, but to me the emphasis on
network position somewhat sweeps under the rug the possibility that the patterns might
be driven by something simpler, e.g. overall sociability of individuals. I also wonder
whether other unmeasured factors, such as spatial positioning in the experimental
arena, could be driving these patterns. To its credit, the study does test for the effect of
overall activity level. However, I think these potentially confounding factors and
subtleties about network position vs sociability should be mentioned in the discussion.
In addition, while proximity is often used as a proxy for social interaction in animal social
networks studies, it does not necessarily imply it and this limitation should also be
discussed.

*It is possible that some network position measures can be explained by some simpler*
*underlying mechanism (e.g. sociability, instrength, outstrength, activity, etc.). However,*
*the strength of network analysis is that while simpler mechanisms may explain some of*
*the variation in network positions, there is still important variation that can only be*
*explained through the descriptive power of network analysis. As you correctly noted,*
*while instrength and outstrength are somewhat correlated with other network position*
*measures, there is much variation left unexplained by instrength/outstrength alone*
*(Supplementary Fig. 2). Part of this is likely due to the fact that it is less clear how*
*network position measures that encapsulate indirect interactions (e.g. eigenvector*
*centrality, betweenness centrality, and clustering coefficient) can be directly tied to*
*simpler mechanisms attributable to a focal individual (e.g. sociability,*
*instrength/outstrength, activity, etc.). It is certainly a challenge to deconstruct the full*
*underpinnings of social network structure; whether studying simpler behavioral*
*underpinnings such as sociability and activity, or studying the more fine-grained genetic*
*underpinnings of the genotype-to-social phenotype relationship (see response to*
*Reviewer #1's comment: line numbers 140-154). We've attempted to address some of*
*the mechanistic links from genotype-to-social network position by amending our*
*discussion on lines 296-302. To your last point about how "proximity is a proxy for social*
*interactions [but] does not necessarily imply it," we wholeheartedly agree and have*
*addressed this on lines 401-403.*

Finally, I have some concerns about the statistics. Overall, a large number of statistical
tests have been conducted, increasing the possibility of false positive results, which
does give me some overall cause for concern. For instance, the use of both day 1 and
264 day 2 networks as separate predictors in the models seems not fully justified.

*See response to your following comment.*

On a somewhat related note: are these two days correlated? It would make sense that,
if social network positions are a real phenomenon here, that these would be somewhat
repeatable across days, and this would be worth investigating.

*We went back and tested for Kendall's rank correlations between network position*
*measures taken one and two days after social groups were established. We found that*
*individuals indeed displayed consistency in their network position measures across*

*days. We believe this additional finding imparts an additional strength to the manuscript,*
*so thank you very much for this suggestion! We've incorporated these changes into the*
*manuscript on lines 178-185 and 509-513., and 549-552.*

*More to your concern above about the number of statistical tests conducted and the risk*
*of false positive results: Because the network position measures were correlated across*
*the two days, we now see that presenting separate fitness analyses for network data*
*collected on days 1 vs. 2 is redundant. As such, we decided to analyze how network*
*position measures covaried with fitness traits using only network data collected from the*
*first day, as this was the day with the more robust sample size (lines 549-552). This*
*reduces the number of statistical tests conducted, for models testing how network*
*position and fitness metrics covary, by half. We hope this alleviates yours and potential*
*other readers' concerns about the risk for false positive results.*

*Furthermore, we'd like to note that our application of a Bonferroni correction to our*
*analyses of how fitness and network measures covary is likely hyper-conservative. We*
*applied a Bonferroni correction of five (adjusted significance threshold is $P < 0.01$)*
*because each of our five network position measures was analyzed in its own model*
*(because multicollinearity prevented us from being able to include all five in a single*
*model for each fitness measure). However, Bonferroni corrections are best suited for*
*multiple independent tests. Because our network position measures are correlated and*
*non-independent, this means that our adjusted significance thresholds are likely*
*conservative, increasing the possibility of false negative results. We've included a*
*statement to this effect in the manuscript (line numbers 570-572).*

*Secondly, I have some concerns about the permutation tests described in the methods.*
*It would be nice to have a bit more explanation of what exactly was done here. From my*
*understanding, the authors assessed significance of all the effects by comparing the*
*test statistic associated with each metric to the distribution of test statistics for that*
*metric in models where individual identity labels had been permuted within groups (?).*
*However, it would seem that this randomisation might not make sense for all*
*comparisons. For instance, for looking at the effect of the random factor genotype while*
*controlling for sex, a reasonable randomisation would be to permute the labels within*
*each sex separately (rather than mixing them). Also, I wondered about the possible*
*effects of individual consistency (i.e. consistency not within a genotype but within a*
*single individual in network position) that could be confounded here, since individual ID*
*is (understandably) not included as a random effect but network measurements for each*
*individual have been replicated twice (therefore genotype ID might pick up some of the*
*variation that actually should be attributed to individual ID here).*

*Your understanding of how we conducted our permutation tests is correct. We assessed*
*the significance of sex, sex-by-environment interactions, genotype, and genotype-by-*
*environment interactions by comparing the test statistics from the observed models to*
*1000 null model test statistics in which the variables of interest (i.e. sex and genotype)*
*were permuted within each social network.*

*To your point about assessing the significance of genotype and genotype-by-*
*environment interactions by randomising genotype within sex within each social network*
*(as opposed to randomising genotype within each social network as we initially did in*
*the manuscript), we agree with your assessment that this is a more reasonable*
*randomisation and have adjusted our models and results accordingly. Doing so did not*
*qualitatively change any of our results, with the exception of we now see a significant*
*genotype-by-environment interaction for the network position of clustering coefficient.*
*We've clarified how we conducted the permutation tests on lines 490-497.*

*To your point about individual consistency and how genotype ID might pick up some*
*variation attributable to individuals: while individual ID was not included as a random*
*factor in models for instrength, outstrength, clustering coefficient, or betweenness*
*centrality; social group ID was included as a random factor in these models. The*
*variation attributable to individual consistency across days is likely to be picked up by*
*the random factor of social group ID, as this random factor indicates which groups (and*
*correspondingly, the individuals within them) were measured twice.*

To address some of these concerns, I suggest that an additional, simpler analysis
addressing the main question of whether social network position is heritable be
conducted with the existing data, as a sanity check. This test would make use of the fact
that social groups have been completely replicated multiple times in the same
conditions (a powerful feature of this study!), and simply check whether the network
metrics associated with different genotypes are repeatable WITHIN each experimental
condition, controlling for sex. In other words, can you show directly that when you don't
change the environment, individuals of the same genotype are more similar in their
network positions than individuals of different genotypes? This could be done by
computing some test statistic associated with a given metric (e.g. the sum of the
variation in instrength across individuals of the same genotype, within a particular
environmental condition), then permuting identities within each social group amongst
individuals of the same sex to generate a null distribution of this test statistic. This would
avoid the need for complicated regression models to address that central question, and
in my view would strengthen the results and better make use of the replicated social
groups.

*If we understand your comments correctly, you're suggesting we analyze the effect of*
*genotype on each network position measure separately for each of the environmental*
*conditions (i.e. nutritional environments) as a way of assessing if "genotypes are*
*repeatable within each experimental condition, controlling for sex." Also, that analyzing*
*the effect of genotype "within a particular environmental condition" would "better make*
*use of the replicated social groups." There are multiple concerns we have about this*
*approach which we have detailed below:*

- *First, analyzing the effect of genotype separately for each of our environmental*
*conditions would split our full sample size amongst five separate analyses (one*
*for each of our five nutritional environments). This massive reduction in sample*
*size for each analysis would greatly reduce the power of our experimental design*
*of replicate social groups, compared to analyzing our data in a single unified*

*model as we've done in this manuscript. Additionally, performing five separate*
*analyses (one for each nutritional environment) instead of a single unified model*
*increases the number of independent statistical tests performed 5x.*

- - *Secondly, the point raised about assessing whether the effect of genotype is*
*repeatable within each environmental condition is accomplished with the*
*multilevel regression models that we utilize. In all of our models assessing the*
*significance of genotype and the heritability of each network position, the*
*nutritional environment was included as a factor in these models. The*
*significance of genotype for each network position measure then indicates this is*
*a consistent phenomenon, regardless of the environmental conditions the social*
*groups experienced.*
- - *Lastly, if we were to analyze the effect of genotype on each network position*
*measure separately for each of the environmental conditions, and we found*
*genotypes to be repeatable in some conditions (but not others), this would be a*
*qualitative result. Discrepancies in results between the treatment conditions*
*could be due to reduced or uneven sample sizes, as opposed to real biological*
*differences. Quantitatively addressing whether the effect of genotype is*
*consistent or variable across environmental conditions would involve testing for a*
*genotype-by-environment interaction effect in a unified model incorporating all*
*environmental conditions, which is what we accomplished in our multilevel*
*regression models.*

*While using multilevel regression models is a more complex approach to addressing*
*how genotype influences social network position measures, this complexity pays off in*
*its ability to more fully utilize the power of our replicated social groups (Bolker et al.*
*2009, Miller et al. 2020). We also believe this approach makes our results more relevant*
*for systems in the wild, where environmental variation is prevalent but cannot always be*
*accounted for in as controlled of a setting as our experimental design. Taken together,*
*we believe our approach of using multilevel regression models is most appropriate for*
*this manuscript.*

- - *Bolker, B. M. et al. Generalized linear mixed models: a practical guide for*
*ecology and evolution. Trends Ecol. Evol. 24, 127–135 (2009).*
- - *Miller, M. L., Roe, D. J., Hu, C. & Bell, M. L. Power difference in a χ^2 test vs*
*generalized linear mixed model in the presence of missing data- A simulation*
*study. BMC Med. Res. Methodol. 20, 1–12 (2020).*

Additional comments:

Figure 1 & 2: The wording of what the box plots show is a little unclear - what is meant
by +/- 1.5x IQR? Also, how far do the lines extend - presumably 95% range?

*The whiskers of the boxplots in figures 1 and 2 extend to the largest/smallest value no*
*further than +/- 1.5x the interquartile range. The interquartile range refers to the*
*difference between the first and third quartiles, which is the box part of the boxplot.*
*We've clarified this in the figure legends for figures 1 and 2.*

L411-413: More clear definitions of what was extracted from the data are needed here.
What was the edge definition? Or, in the case of instrength and outstrength, what is the

metric computed - fraction of time being interacted with / interacting with others? Also,
definitions of the other 3 network metrics should be given, since these can be computed
slightly different ways.

*An edge was defined as the total duration of time any pairwise combination of flies*
*spent interacting throughout the duration of a video. We've clarified this in the*
*manuscript on lines 404-405. For the definitions of our five network position measures,*
*we originally chose to define them in the body of the manuscript (lines 90-99). We now*
*see that defining them in the methods as well is prudent. We've now done so on lines*
*408-417.*

L449: Would this not induce a bias in the data, since males that never mated are not
included in the latency analysis? Perhaps one could use survival analysis with right-
censored data to get around this issue?

*Thank you for this suggestion. We've amended our analyses of male mating latency to*
*now use Cox proportional-hazards models, which allow us to right censor our data and*
*include males who were never observed to mate in our analyses. Interestingly, doing so*
*now makes our results for male mating latency more similar to our results for total*
*number of male matings. Specifically, males with high instrength, outstrength, and*
*eigenvector centrality tended to mate more quickly (low mating latency) on nutritional*
*environments that were either high-calorie or carbohydrate-rich, but took longer to mate*
*(high mating latency) on nutritional environments that were low-calorie or protein-rich.*
*This finding adds to our evidence for context-dependent selection operating on social*
*network positions. These changes can be found in the manuscript on lines 204-205,*
*214-216, 231-235, 238-241, 450-454, and 528-535.*

Methods - Replication: There is quite a lot of missing data in this study, and a casual
reader might not notice this. Could this be mentioned in the results and/or discussion,
as a possible limitation? The information in this section could also be summarised in a
table, which might aid the reader.

*We've incorporated a table summarizing our sample sizes (Supplementary Table 4) per*
*your suggestion. Keeping a fly alive long enough to gather behavioral data can*
*sometimes be difficult. Keeping all 20 flies alive in our social groups long enough to*
*video them was a very difficult endeavor. If so much as a single fly died within our social*
*groups of 20 flies, the social group was no longer a replicate in the sense that there was*
*variation in the number and composition of genotypes present. This indeed led to a lot*
*of missing data, which we now discuss on lines 440-444.*

L442-443: Since there were only 56 fully tracked videos in the end (out of a possible
$98 \times 2 = 196$), how many of these were independent groups (vs replicates of the same
group)?

*Forty-three (43) of the 56 fully tracked videos were from independent groups. Of these,*
*13 groups had replicate measures of network structure taken across both days of*

*videoing. This information is now outlined in Supplementary Table 4, and on lines 445-*
*446.*

Reviewer #3 (Remarks to the Author):

In this very interesting paper, the Authors conduct a genotypically controlled assay on
heritability and effect of environmental factors (nutrients available) of social network
features. They show that multiple measures of network position are inheritable and that
sex interacts with the environment in determining the relation between fitness and social
network measures. I enjoyed reading the paper. There are a few points that I believe
should be clarified and addressed.

*Thank you very much for your review! We're glad to hear you found our manuscript*
*interesting and enjoyed reading it.*

Would it be worth to quantify to which extent genotype explains variability in social
network position?

*The extent to which genotype explains variation in a phenotype is defined as the broad-*
*sense heritability of said phenotype. We've clarified this on lines 120-123.*

The fact that one crossing and F1 is representative of a natural population should be
justified/discussed.

*Thank you for the suggestion. We've clarified this on lines 343-345 and 351-354.*

69: related to the point above: they are not really identical

*We're not entirely sure what you mean by this comment. The parental generation of*
*inbred genotypes is homozygous across the vast majority of loci in their genome. Thus,*
*any meiotic recombination that happens between their chromosomal pairs will result in*
*similarly identical gametes. When an individual of one inbred homozygous genotype is*
*then mated with an individual of a different inbred homozygous genotype, each of the*
*inbred genotypes passes on the same gamete, every time that mating cross occurs*
*(provided the sex of each inbred homozygous genotype is additionally kept constant to*
*control for the inheritance of sex chromosomes). The resulting F1 progeny from the two*
*inbred homozygous genotypes is then heterozygous at every loci their parents differed*
*in. All F1 progeny of a given sex are also genetically identical to each other, because*
*each inherited the same genes from their inbred homozygous parents. Because inbred*
*homozygous lines of flies can be maintained as stocks over multiple generations, where*
*flies of each genotype only mate with flies of their same genotype, we can create a*
*mating cross of two inbred genotypes at any point in time, creating genetically identical*
*F1 individuals. In our experiment, we did this with 40 inbred homozygous genotypes*
*and 20 unique mating crosses, to create 20 heterozygous individuals, one of each of*
*which was included in a single social group. Each social group contained the same 20*
*heterozygous genotypes derived from the same 40 inbred homozygous genotypes,*
*meaning that each social group was a genetically identical replicate. We discuss how*
*our replicate social groups were created in more detail on lines 343-367 and*
*Supplementary Fig. 1.*

13 and in the manuscript: you refer to multiple measures of social networks... this is a
bit vague. Can you focus on those that are independent from each other?

*Because measures of social network position are derived from the same underlying*
*network of social interactions, measures of social network positions are inherently non-*
*independent. Similarly, they are also often highly correlated, as exemplified in*
*Supplementary Fig. 2. There is however, important variation between measures of*
*social network positions that isn't entirely explained by their non-independence to one*
*another. For example, while instrength and outstrength were the two most highly*
*correlated measures of social network position in our study, there was still important*
*variation between them explained by the sex of individuals (Fig. 1). We've clarified how*
*measures of network position are non-independent and what this means for the*
*evolution of network structure on lines 321-324.*

134-138, 142-143: these experiments have been conducted on very different species, it
might be a misleading comparison.

*Estimates of heritability for various phenotypes are indeed likely to vary between very*
*different species (and between different populations or different social groups for that*
*matter). Because there is so little published evidence of whether social network*
*positions are heritable, we find that framing our results in the context of the few studies*
*that do exist is a prudent approach to understanding whether the heritability and*
*evolutionary potential of social network structure is a widespread phenomenon or highly*
*variable. We've sought to clarify this comparison by amending our discussion on lines*
*316-321.*

Fig 3: datapoints should be visualised

*We've added datapoints to Fig. 3.*

225: aren't social pressures a kind of selective pressure? it seems you argue that
natural selection might work using different principles

*Social pressures are absolutely a kind of selective pressure. How social interactions*
*affect selection is perhaps best explained by social selection theory (Moore et al. 1997,*
*Bijma et al. 2007, and McGlothlin et al. 2010), which is what we meant by the statement*
*you've pointed out. We've clarified this statement on lines 338-340.*

- *Moore, A. J., Brodie III, E. D. & Wolf, J. B. Interacting phenotypes and the*
*evolutionary process: I. Direct and indirect genetic effects of social interactions.*
*Evolution 51, 1352– 1362 (1997).*

- *Bijma, P., Muir, W. M., Ellen, E. D., Wolf, J. B. & Van Arendonk, J. A. M.*
*Multilevel selection 2: Estimating the genetic parameters determining inheritance*
*and response to selection. Genetics 175, 289–299 (2007).*

- McGlothlin, J. W., Moore, A. J., Wolf, J. B. & Brodie, E. D. *Interacting phenotypes*
*and the evolutionary process. III. Social evolution. Evolution 64, 2558–2574*
*(2010).*

extended data figure 2:

- I wonder whether readability can be improved for the scatterplots

*We made the datapoints in Supplementary Fig. 2 scatterplots translucent to better*
*visualize the density of their distributions.*

Minor

5-6 why not mentioning the traits investigated in this paper?

*Thank you for the suggestion. Due to space constraints, we've now removed references*
*to fitness traits we did not measure in the abstract.*

19-22 can this part be more clear on your findings? (e.g. referring to 218-219)

*Due to space constraints in the abstract, we decided to remove the sentence in*
*question.*

24: what do you mean with "confounds"?

*We've removed this word for clarity (line number 35).*

REVIEWER COMMENTS

Reviewer #1 (Remarks to the Author):

Well done to the authors on a thorough and convincing job responding to the reviewer comments.

I believe I didn't explain myself clearly with this comment

"It would be helpful to have the link (or lack thereof) from genotype to network position to context spelled out in clearer terms as this is an important part of the results." I wasn't referring to the G x phenotype relationship but the G x phenotype x context (nutritional manipulation). I thought a bit more could be said in the text about the absence of significant result there as these results everything together in an important way. However, it appears that adjusted analyses have found a significant result with respect to clustering coefficient and so the discussion has been nicely amended to reflect that.

Otherwise, I have nothing additional to add!

Reviewer #2 (Remarks to the Author):

The authors have done a nice job of addressing both my comments and those of the other reviewers. In particular, I am glad to see the various statistical concerns were addressed, and parts of the text clarified and limitations noted.

I wanted to clarify about my earlier suggestion regarding a simple test of whether flies with the same genotype in the same environment are more similar than those with different genotypes in a given environment. The authors rightly objected to the idea of carrying out 5x as many statistical tests! However, this was not actually what I meant, so I will try to explain more clearly below. I am leaving this here only as a suggestion, since the authors have already made clear why they chose to use a multilevel regression approach. Also to be clear, this would not replace the multilevel regression, it would be in addition.

My suggestion effectively amounts to measuring the repeatability of network structure (or node-level properties) when you have the same (genetically) individual in the same environment. For simplicity, imagine that you had two groups of 20 in two environments. What we might be interested to know is whether, if you compare a fly in a given environment of genotype A to its "partner" in that same environment in the other replicate, is this more similar than if you compared it to a fly of a different genotype B in the same environment. To get at this, one could compute some test statistic of those pairs that represents their similarity (for instance, the correlation or the intra-class correlation coefficient, or my earlier suggestion), across the entire dataset (including both environments - I think this was the part that was unclear!). Then, one could randomize the genotypes (shuffling within sex, and within each environment separately) and compute this test statistic again. Repeat this randomization 1000 x to generate a null distribution and see whether your real correlation / ICC is significantly higher than this

null distribution. This approach would actually be similar in spirit to the full regression approach, but here you are not trying to estimate the contribution of the environment but rather incorporating the environment effect into the null model to really focus on the repeatability of network position of a given genotype under "identical" conditions.

In any case, as I mentioned earlier I am not going to insist that the authors carry out this analysis, as I find the revisions sufficient as is. I am only mentioning it because I see this as a unique opportunity to show, without any complicated models involved and under a well-controlled scenario, that network properties are heritable.

Reviewer #3 (Remarks to the Author):

Thanks to the Authors for addressing a few issues raised.

- Let me clarify the previous point about isogenic lines (I was too synthetic). There are a few reasons why if we capture a population of flies and use them to build isogenic lines, and then we cross the isogenic lines, we won't have a new population indistinguishable from the original one:

- 1) many alleles will be lost because recessive lethal in general
- 2) many alleles will be lost because recessive lethal in the specific lab environment
- 3) linkage disequilibrium, particularly relevant in species with inversions (it applies to this species)

It would be worth specifying that your gene pool and your flies are not exactly identical to a natural population. This is also because linkage disequilibrium (and the presence of inversions in the flies' genomes) can make it difficult to disentangle between traits. In natural populations, some traits might be more independent than in the first generations of crosses from isogenic lines.

- All reviewers (in particular Rev 1 and 3) pointed at collinearity and non-independence between measures related to the social networks. You have now specified that these measures are not independent. Have you tried using a mediator/moderator analysis approach to better understand the role of different variables?

- A major challenge for a paper that argues about "Selection on heritable social network positions" is to understand whether social network position is selected for, or whether it simply correlates with other traits that are under selection, that influence social network as a side-effect. Would it be possible to move beyond correlations and experimentally manipulate the position in the social network, and then see whether changing or disrupting the social network position influences fitness?

- An important point here is to exclude that simpler traits (e.g. size of the fly) are the major (or unique) drivers of the effects, and that individual traits (that can influence the position in a social network), rather than social network per se are the object of selection. I am wondering whether an experiment in which you use crossed flies and then used only those flies of that cross with a certain phenotype (e.g. small or big size) can help you to estimate whether a simple individual trait is a major or minor driver of

the change.

- If the Authors cannot causative link social network position with selection, I would suggest that they limit their claims to the level of an "association" or correlation.

Dear Colleagues,

Thank you very much for your second round of comments on our manuscript, "Selection
on heritable social network positions is context-dependent in *Drosophila melanogaster*."
In response to your comments, we have:

- • Further clarified and justified key aspects of our experimental approach
 - • More clearly communicated our statistical results and their interpretation
 - • Completed additional edits for clarity throughout the manuscript

We believe these changes have further strengthened the findings of this manuscript,
and greatly improved its readability for a broad audience. Please see below our point-
by-point response to reviewers' comments. Our responses are italicized and written
below each reviewer point. We've included a revised version of the manuscript with
changes indicated by the Track Changes function in Microsoft Word. Thank you for your
time spent reviewing our work. We hope that, with any additional changes you suggest,
our manuscript may soon be published in *Nature Communications*.

Thank you,
Eric Wice
Corresponding author
eric.wesley.wice@gmail.com

**REVIEWER COMMENTS**

Reviewer #1 (Remarks to the Author):

Well done to the authors on a thorough and convincing job responding to the reviewer
comments.

I believe I didn't explain myself clearly with this comment
"It would be helpful to have the link (or lack thereof) from genotype to network position
to context spelled out in clearer terms as this is an important part of the results." I wasn't
referring to the G x phenotype relationship but the G x phenotype x context (nutritional
manipulation). I thought a bit more could be said in the text about the absence of
significant result there as these results everything together in an important way.
However, it appears that adjusted analyses have found a significant result with respect
to clustering coefficient and so the discussion has been nicely amended to reflect that.

Otherwise, I have nothing additional to add!

*Thank you!*

Reviewer #2 (Remarks to the Author):

The authors have done a nice job of addressing both my comments and those of the
other reviewers. In particular, I am glad to see the various statistical concerns were
addressed, and parts of the text clarified and limitations noted.

I wanted to clarify about my earlier suggestion regarding a simple test of whether flies
with the same genotype in the same environment are more similar than those with
different genotypes in a given environment. The authors rightly objected to the idea of
carrying out 5x as many statistical tests! However, this was not actually what I meant,
so I will try to explain more clearly below. I am leaving this here only as a suggestion,
since the authors have already made clear why they chose to use a multilevel
regression approach. Also to be clear, this would not replace the multilevel regression, it
would be in addition.

My suggestion effectively amounts to measuring the repeatability of network structure
(or node-level properties) when you have the same (genetically) individual in the same
environment. For simplicity, imagine that you had two groups of 20 in two environments.
What we might be interested to know is whether, if you compare a fly in a given
environment of genotype A to its "partner" in that same environment in the other
replicate, is this more similar than if you compared it to a fly of a different genotype B in
the same environment. To get at this, one could compute some test statistic of those
pairs that represents their similarity (for instance, the correlation or the intra-class
correlation coefficient, or my earlier suggestion), across the entire dataset (including
both environments - I think this was the part that was unclear!). Then, one could
randomize the genotypes (shuffling within sex, and within each environment separately)
and compute this test statistic again. Repeat this randomization 1000 x to generate a
null distribution and see whether your real correlation / ICC is significantly higher than
this null distribution. This approach would actually be similar in spirit to the full
regression approach, but here you are not trying to estimate the contribution of the
environment but rather incorporating the environment effect into the null model to really
focus on the repeatability of network position of a given genotype under "identical"
conditions.

In any case, as I mentioned earlier I am not going to insist that the authors carry out this
analysis, as I find the revisions sufficient as is. I am only mentioning it because I see
this as a unique opportunity to show, without any complicated models involved and
under a well-controlled scenario, that network properties are heritable.

*Thank you very much for this suggestion! In the future we will more fully consider how*
*different types of simulations and permutations may provide unique insights into the*
*structure of social network data. Since the current manuscript is already quite dense*
*with analyses, and the existing analyses are both robust and well-suited to the current*
*goals, we agree that no changes are necessary at this time.*

Reviewer #3 (Remarks to the Author):

Thanks to the Authors for addressing a few issues raised.

- Let me clarify the previous point about isogenic lines (I was too synthetic). There are a
few reasons why if we capture a population of flies and use them to build isogenic lines,
and then we cross the isogenic lines, we won't have a new population indistinguishable
from the original one:

1) many alleles will be lost because recessive lethal in general

2) many alleles will be lost because recessive lethal in the specific lab environment

3) linkage disequilibrium, particularly relevant in species with inversions (it applies to
this species)

It would be worth specifying that your gene pool and your flies are not exactly identical
to a natural population. This is also because linkage disequilibrium (and the presence of
inversions in the flies's genomes) can make it difficult to disentangle between traits. In
natural populations, some traits might be more independent than in the first generations
of crosses from isogenic lines.

*Thank you, it is true that no collection of inbred lines can perfectly represent a wild*
*population (although we note that we did not claim that our lines are genetically*
*indistinguishable from the original population). For nearly all traits, the genotype-to-*
*phenotype relationship is complex, with many different functional mechanisms at*
*different levels of organization influencing the relationship between genotype and social*
*network position (e.g. genotype-to-gene expression, gene expression-to-protein*
*expression, protein expression-to-individual behavior, etc.). Thus, understanding the*
*many mechanisms translating genotype to social network position is outside the scope*
*of this manuscript, as our goal was to establish the evolvability of network structure and*
*how the nutritional environment contributes to shaping the evolution of social group*
*structure.*

*Given the available information about how genetic variation produces variation in social*
*behavior, it is most likely that the many mechanisms underlying variation in each aspect*
*of social network position are highly polygenic, with few, if any major effect loci (e.g.,*
*see Matt Rockman's paper, "The QTN Program and the alleles that matter for*
*evolution: all that is gold does not glitter. Evolution 2011,66(1): 1-17). If, in the future,*
*the genetic basis of social network position is dissected in this population, and a major-*
*effect allele influencing some aspect(s) of social network position is segregating in an*
*inversion, we would consider this to be a surprising and very exciting discovery that*
*does not undermine the findings reported in the current manuscript. From an*
*evolutionary perspective, the most important parameters are heritability and the*
*covariance between traits and fitness, which we have presented here.*

*In response to your comments, we added a statement to the methods section justifying*
*our approach of using inbred lines as a balance between our experimental need to*
*create replicated social groups with our desire to produce evolutionarily-relevant*
*inferences about heritability and selection (lines 310-312).*

- All reviewers (in particular Rev 1 and 3) pointed at collinearity and non-independence
between measures related to the social networks. You have now specified that these
measures are not independent. Have you tried using a mediator/moderator analysis
approach to better understand the role of different variables?

*While many of our network position measures are colinear (see Supplementary Fig. 2),*
*there is still important and distinct variation between them. This is why we included a*
*correction for multiple testing, because there is important variation between the network*
*position measures that isn't explained by their multicollinearity. For example, while*
*instrength and outstrength were highly correlated, there was still important variation*
*between these measures explained by the sex of the flies (Fig. 1).*

*From a more practical perspective, we are unaware of any implementations of*
*mediation analysis that can accommodate the complex error structures of the sort*
*recorded for our social network data (described in the section Analyses of Social*
*Network Positions). In this context, and with the other reviewers' suggestions that the*
*paper is already quite dense with analyses, we are unsure about what additional*
*mediation analysis would add.*

*We describe our ideas about genetic correlations more broadly in our answers to your*
*other comments. We also describe the evolutionary implications of the non-*
*independence of social network traits in the manuscript in lines 286-291.*

- A major challenge for a paper that argues about "Selection on heritable social network
positions" is to understand whether social network position is selected for, or whether it
simply correlates with other traits that are under selection, that influence social network
as a side-effect. Would it be possible to move beyond correlations and experimentally
manipulate the position in the social network, and then see whether changing or
disrupting the social network position influences fitness?

*The potential for unmeasured, but important, covariates is a problem for all selection*
*analyses, and indeed, most analyses in most sciences. Measuring behavior, fitness,*
*and their genetic covariance under highly-controlled conditions, as we have done here,*
*is one method for reducing, as much as possible, the dimensionality of potential*
*confounding factors. Furthermore, the genetic covariance between traits and fitness is*
*still relevant to predicted evolutionary change even if this relationship is mediated,*
*partially or fully, by variation in other traits (with some assumptions).*

*Our finding that each social network measure contributes to fitness outcomes in unique*
*ways (including some that don't seem to contribute at all) is consistent with our*
*expectations that different aspects of social network position, as partially-distinct traits,*
*affect fitness via their influence on social interactions. This is not what we would expect*
*if both social network position and fitness measures were explained by a single*
*underlying covariate, e.g., "general health." As we note above, understanding the*
*underlying mechanisms that together produce social network position and its variation*

would not alter the core conclusions of the current manuscript, i.e., that social network
position is heritable and selection on network position is context-dependent.

*As far as we know, there is no available system to manipulate an animal's social*
*network position without changing anything else (e.g., the animal's genotype or the*
*environment in which the test is conducted, etc). One way we are working on this*
*problem currently is to manipulate the genotypic composition of our social groups to*
*identify whether genotypes maintain consistent social network position and fitness*
*across different group types. As you can imagine, this is a major additional undertaking*
*and cannot be adequately included in the current manuscript.*

- An important point here is to exclude that simpler traits (e.g. size of the fly) are the
major (or unique) drives of the effects, and that individual traits (that can influence the
position in a social network), rather than social network per se are the object of
selection.

*Thank you. First, we note that the size of the fly alone could not directly influence our*
*measures of social interactions, because the radius in which flies were considered to be*
*interacting was fixed for all individuals. In other words, bigger flies (i.e., females) did not*
*have a larger spatial area in which social interactions were counted. We added a*
*sentence to clarify this in lines 353-356.*

*Further, we have included activity as a covariate in our analyses, and this did little to*
*explain our findings that social network traits are heritable. At a deeper level, it is*
*interesting to consider what, if anything, is the difference between "traits that lead*
*individuals to have a particular social network position" being under selection versus*
*social network position as under selection "per se"? For example, would a trait like*
*fleeing from a predator not be considered "under selection per se" if that trait were*
*mediated in part by metabolic rate and hormone titers? In addition, there is fascinating*
*evidence emerging from human psychiatric genetics illustrating that measurements of*
*seemingly underlying physiological phenotypes, such as brain region activation, do not*
*necessarily get us "closer to the genes," compared to directly studying the emergent*
*phenomenon (e.g., schizophrenia) itself (e.g., see Flint & Manufo, "The endophenotype*
*concept in psychiatric genetics," Psychological Medicine 2006, 37(2): 163-180).*
*Although most of this discussion is beyond the scope of the current manuscript, we*
*provide a short overview of some recent findings on the underlying genetic basis of*
*social behaviors, and the need for more research in this area, on lines 263-269.*

*Honestly, it would make our lives much easier if there were a single, easily-measurable*
*trait, like body size, that adequately explained variation in social network position and its*
*fitness consequences! That would make our work much, much easier. However, the*
*evidence available does not currently support this interpretation.*

I am wondering whether an experiment in which you use crossed flies and then used
only those flies of that cross with a certain phenotype (e.g. small or big size) can help

you to estimate whether a simple individual trait is a major or minor driver of the change.

*It would be fascinating to understand how early-life plasticity expressed within*
*genotypes may contribute to individual variation in network position, as suggested here.*

*This type of experiment would be a major undertaking and would address distinct*
*questions from the ones posed in the current manuscript.*

- If the Authors cannot causative link social network position with selection, I would
suggest that they limit their claims to the level of an "association" or correlation.

*Thank you for this suggestion. We have edited our language throughout the manuscript*
*to better communicate this point.*

REVIEWERS' COMMENTS

Reviewer #3 (Remarks to the Author):

Thanks for addressing a few issues and further clarifying the manuscript.